# Long-term alterations of collagen reconstruction and basement membrane regeneration after corneal full-thickness penetrating injury in rabbits

**Jingjing Chen**[◐][¤]**, Yuqing Luo**[◐][¤]**, Luting Xie**[¤]**, Na Meng**[¤]**, Sumei Li**[¤]**, Shifang Xiao**[¤]**, Xia Li**[¤]*

Department of Ophthalmology, the First Affiliated Hospital of Guangxi Medical University, Nanning, Guangxi, China

◐ These authors contributed equally to this work.
¤Current address: Department of Ophthalmology, the First Affiliated Hospital of Guangxi Medical University, Nanning, Guangxi, China
* lixiagmu066@163.com

## Abstract

### Purpose

To investigate the long-term alterations of collagen reconstruction and basement membrane (BM) regeneration after corneal full-thickness penetrating injury in rabbits.

### Methods

The corneal full-thickness penetrating injury model was established in the left eye of New Zealand White rabbits using a 2.0 mm trephine. All corneas were evaluated using slit-lamp photography, hematoxylin and eosin staining, immunofluorescent staining for collagen types I and III (Col I, III), and transmission electron microscopy for collagen fibers and basement membrane.

### Results

Between 3 days and 3 weeks, Col I and III expression were documented, exhibiting a largely disorganized distribution throughout the stromal thickness. At 3 weeks, the epithelial basement membrane (EBM) partially regenerated. From 3 weeks to 2 months, Col III was undetectable in the anterior stroma but present in the posterior stroma; Col I was disorganized in the posterior stroma. At 2 months, Descemet's membrane (DM) exhibited incomplete regeneration. From 3 to 4 months, Col I was disorganized in only a small part of the posterior stroma; Col III persisted in the posterior stroma; the EBM fully regenerated while DM exhibited incomplete regeneration.

### Conclusions

Following full-thickness corneal injury, persistent fibrosis within the posterior stroma appears to be primarily responsible for the persistence of corneal scarring. Notably,

**Data availability statement:** The data supporting this study are available from the Biostudies database (DOI 10.6019/S-BSST1805).

**Funding:** This work was supported by the National Natural Science Foundation of China Grant number 81,360,144, China) and the Natural Science Foundation of Guangxi Province (Grant number 2017GXNSFAA198250, Guangxi, China).

**Competing interests:** The authors have declared that no competing interests exist.

regeneration of the EBM coincides with remodeling of the anterior stroma, whereas incomplete regeneration of DM is associated with posterior stromal fibrosis.

## Introduction

Corneal injury can lead to scar formation, characterized by altered collagen fiber properties (diameter, arrangement, and orientation) that compromise corneal transparency [1–3]. During repair, the corneal stroma exhibits partial self-reconstruction involving keratocyte and extracellular matrix (ECM) remodeling [4–10]. However, the factors primarily responsible for the pathological regression of scar formation remain inadequately understood. Our previous study demonstrated the persistence of alpha-smooth muscle actin (α-SMA), a myofibroblast marker, in the posterior stroma after corneal perforation [11]. Corneal full-thickness penetrating injury is a common ocular trauma [12]. Despite suggestions that the stromal cells return to a quiescent state accompanied by the basement membrane (BM) re-establishment [13], the relative contributions of the anterior and posterior corneas remain unclear. Therefore, this study aimed to investigate the long-term dynamic changes in collagen reconstruction and BM regeneration following full-thickness corneal injury.

The cornea contains two critical basement membranes: the epithelial basement membrane (EBM), which underlies the corneal epithelium [14–19], and Descemet's membrane (DM), located beneath the corneal endothelium [20,21]. Both EBM and DM are pivotal in the pathophysiology of stromal fibrosis, serving as structural and functional barriers that regulate the influx of cytokines and growth factors (GFs) into the injured corneal stroma [22,23]. Damage to EBM and DM facilitates the entry of substantial amounts of transforming growth factor-beta (TGF-β) from the corneal epithelium, tears, and aqueous humor into the stroma, along with other growth factors such as platelet-derived growth factor (PDGF). This cascade initiates the differentiation of precursor cells into myofibroblasts [14,20,24]. If basement membrane injury persists beyond three weeks, elevated concentrations of TGF-β and PDGF permeate the stroma, inducing keratocytes to transform into corneal fibroblasts. These fibroblasts, along with fibrocytes, further differentiate into mature myofibroblasts. The newly formed myofibroblasts synthesize an ECM containing collagen and other components, which plays a critical role in repairing corneal wounds [14]. However, research on the repair process of full-thickness stroma after a corneal penetrating injury remains limited.

Given the protracted nature of corneal wound healing, this study investigated long-term alterations in stromal collagen reconstruction and basement membrane regeneration for up to 4 months following full-thickness penetrating injury in rabbits. This systematic analysis aimed to elucidate long-term changes in collagen and basement membrane remodeling, with the ultimate goal of informing the development of novel interventions to prevent or minimize scar formation during corneal wound healing.

## Materials and methods

Male New Zealand White rabbits (12–15 weeks old, 1.5–2.0 kg) were obtained from the Experimental Animal Center of Guangxi Medical University. All procedures adhered to the Association for Research in Vision and Ophthalmology (ARVO) Statement for the Use of Animals in Ophthalmic and Vision Research (protocol number: 2024E50301), approved by the Ethics Committee of the First Affiliated Hospital of Guangxi Medical University.

## Corneal penetrating injury model

Forty-eight New Zealand White rabbits were randomly assigned to eight groups (n = 6/group) representing different post-operative time points. The left eyes of each rabbit received a penetrating injury using a sterile 2.0 mm trephine (Storz, USA) to create the model, while the right eyes served as controls. The injury model creation followed established protocols [25]. Briefly, each rabbit underwent left eye trephination at the central cornea using the aforementioned sterile trephine. Prior to the procedure, the experimental eyes were dilated with 0.5% compound tropicamide eye drops (SHENYANG XINGQI PHARMACEUTICAL CO, LTD, China) to prevent iris incarceration in the trephined area during the operation. Following pupil dilation in the rabbits, a precise weighing procedure was conducted. General anesthesia was subsequently administered via intravenous infusion of 1% pentobarbital sodium (at a dose of 3ml/kg) through the marginal ear vein. A 0.5% proparcaine hydrochloride ophthalmic anesthetic solution (Alcon-COUVREUR, USA) was then applied to the ocular surface of the operative eye. After confirming the effectiveness of both general and local anesthesia, a pediatric eyelid speculum was carefully positioned over the surgical eye to fully expose the rabbit's cornea. The eye was stabilized using forceps, and a sterile trephine with a 2.0 mm diameter was precisely aligned at the center of the cornea. Using the rotary trephine, the entire layer of corneal tissue was meticulously excised, revealing the aqueous humor. Next, 0.3% Levofloxacin gel was applied to the corneal wound to prevent anterior chamber collapse and iris adhesion. Notably, the rabbits regained consciousness within 30 minutes post-procedure. Within 15 minutes, fibrin clots formed on the corneal wound, effectively sealing it and re-establishing the anterior chamber. The rabbit corneal perforation injury model was successfully established. Postoperatively, one drop of 0.3% Levofloxacin gel was administered every 8 hours for the first three days, followed by 0.5% Levofloxacin eye drops (YABANG Pharma, China) every 8 hours for two weeks. Mitomycin C and topical corticosteroids were not utilized in this study. Any instances of corneal ulceration or neovascularization were excluded from the study.

Standardized broad-beam illumination corneal slit-lamp photographs were obtained at various time points: 3 days, 1 week, 2 weeks, 3 weeks, 1 month, 2 months, 3 months, and 4 months post-surgery. Euthanasia was performed by intravenous injection of pentobarbital sodium 100 mg/kg (AG, Germany). Corneoscleral rims from both wounded and unwounded eyes were excised using forceps and scissors, ensuring no contact with the cornea itself, for subsequent histopathology and corneal ultrastructural examination.

## Slit-lamp microscopic examination

At each time point following corneal injury, all rabbit eyes were evaluated using slit-lamp microscopy. The corneal epithelial defect area was measured using ImageJ software following the instillation of 1% sodium fluorescein solution and photography with cobalt blue light. Corneal opacity was graded on a modified Fantes' scale (0 = no opacity, 4 = complete opacity) as previously described [26,27]. Briefly, grade 0.5 indicates slight opacity with visible pupillary margin and iris details, grade 1 indicates observable opacity with both details visible, grade 2 signifies partial pupillary margin visibility, and grade 3 represents moderate opacity without a visible pupillary margin.

## Histopathologic examination

Histopathological changes in injured and control corneas were assessed using hematoxylin and eosin (H&E) staining. At each designated time point, rabbits were euthanized under deep

anesthesia, and eyes were immediately enucleated. Following fixation in 4% paraformaldehyde (Solarbio, China) for 48 hours, the eyes were paraffin-embedded, sectioned into 4-μm slices, and dewaxed with xylene. These sections were then subjected to H&E staining for histological analysis.

## Immunofluorescence analysis

Immunofluorescence (IF) staining was employed to assess the expression and localization of type I collagen and type III collagen, markers for ECM protein deposition, respectively. The preparation of the wax blocks and the slicing procedures was performed in strict accordance with the previously described methods. Paraffin-embedded corneal sections were subjected to dewaxing in xylene followed by rehydration through a graded ethanol series. After rehydration, the slides were immersed in 3% EDTA and heated in a microwave oven for antigen retrieval. Following natural cooling, the slides were washed three times with 0.1 M phosphate-buffered saline (PBS) and permeabilized with 0.3% Triton X-100 at room temperature for 10 minutes. To minimize non-specific binding, the slides were blocked with 30% normal goat serum in PBS for 1 hour. The slides were then incubated overnight at 4°C with anti-COL1A1 antibody (NB600–450, Mouse anti-Collagen I alpha 1; Novus) at 1:100 dilution in 1% BSA, anti-COL3A1 antibody (sc-271249, Mouse monoclonal antibody, 1:100, Santa Cruz) at 1:30 dilution in 1% bovine serum albumin (BSA). Slides were then incubated for 30 minutes with the secondary antibody (2284614, Alexa Fluor 488 goat anti-mouse antibody; Thermo Fisher Scientific) at dilution of 1:200 (COL1A1), and with the secondary antibody (4408S, Alexa Fluor 488 goat anti-mouse antibody; Cell Signaling Technology) at dilution of 1:50 (COL3A1). Then counterstained with 4',6-diamidino-2-phenylindole (DAPI) (C0065-10ml, Solarbio, China). Tissue sections were visualized using a Leica TCS SP8 microscope (Leica, Wetzlar, Germany) equipped with a Fluoview FV1000 confocal system (Olympus, Japan). Image analysis was performed using ImagePro software (Leica).

## Transmission electron microscopy (TEM)

Corneal tissue processing followed previously established protocols [32]. Briefly, specimens were fixed in a 2.5% glutaraldehyde and 4% paraformaldehyde mixture for 24 hours, followed by incubation in 1% osmium tetroxide for 2 hours at 4°C. Dehydration was achieved through a graded ethanol and acetone series. Subsequently, the central corneal block was embedded in epoxy resin and sectioned using an ultramicrotome (Leica EM UC7, Germany) to 1 μm thickness. Toluidine blue staining facilitated the identification of the corneal injury area. Ultrastructural changes in the corneal tissue during the repair process were observed under a transmission electron microscope (Hitachi H-7650, Japan). Additionally, scanning electron microscopy (TESCAN VEGA3, Czech Republic) was employed to capture micrographs of the tissue.

## Statistical analysis

All data were analyzed using SPSS 17.0 software (IBM-SPSS, Chicago, IL, USA). One-way analysis of variance (ANOVA) was utilized for comparisons among three or more groups. Multiple comparisons between groups were conducted using the LSD-t test. P values < 0.05 were considered statistically significant. All data were presented as the mean ± standard deviation (SD), and all charts were generated using GraphPad Prism 5.0 software.

## Results

### Slit lamp photographs of epithelial defect and corneal opacity in the central cornea

Slit-lamp examination revealed corneal epithelial defect and corneal opacity. As illustrated in Figs 1–3, corneal epithelium displayed complete regeneration by 3 weeks post-injury. Corneal opacity peaked at 1 month and gradually diminished thereafter. The control group exhibited a completely transparent and clear cornea with no fluorescein staining (Figs 1A and 2A). Three days after injury (Figs 1B and 2B), the wound area showed significant fluorescein staining, indicating a substantial epithelial defect. The wound appeared translucent, with visible iris details. One week post-injury (Figs 1C and 2C), the area of fluorescein staining in the wound decreased, and epithelial regeneration progressed toward the wound margins. Peripheral haze was apparent, but the iris remained visible. Two weeks after injury (Figs 1D and 2D), only a small area within the wound displayed fluorescein staining, indicating improved epithelial healing. The wound area exhibited uniform opacity, with a faintly visible iris. By 3 weeks post-injury (Figs 1E and 2E), fluorescein staining was absent, and complete epithelial regeneration was observed. However, the wound area displayed dense opacity, partially obscuring the iris and lens. At one month post-injury (Figs 1F and 2F), the cornea remained free of fluorescein staining with complete epithelial regeneration. The wound area presented complete opacification. Corneal opacity remained dense at 2 months (Fig 2G), although reduced compared to 1 month, with moderate iris and lens obscuration. At three months (Fig 2H), the wound area appeared translucent with a faintly visible iris. By 4 months post-injury (Fig

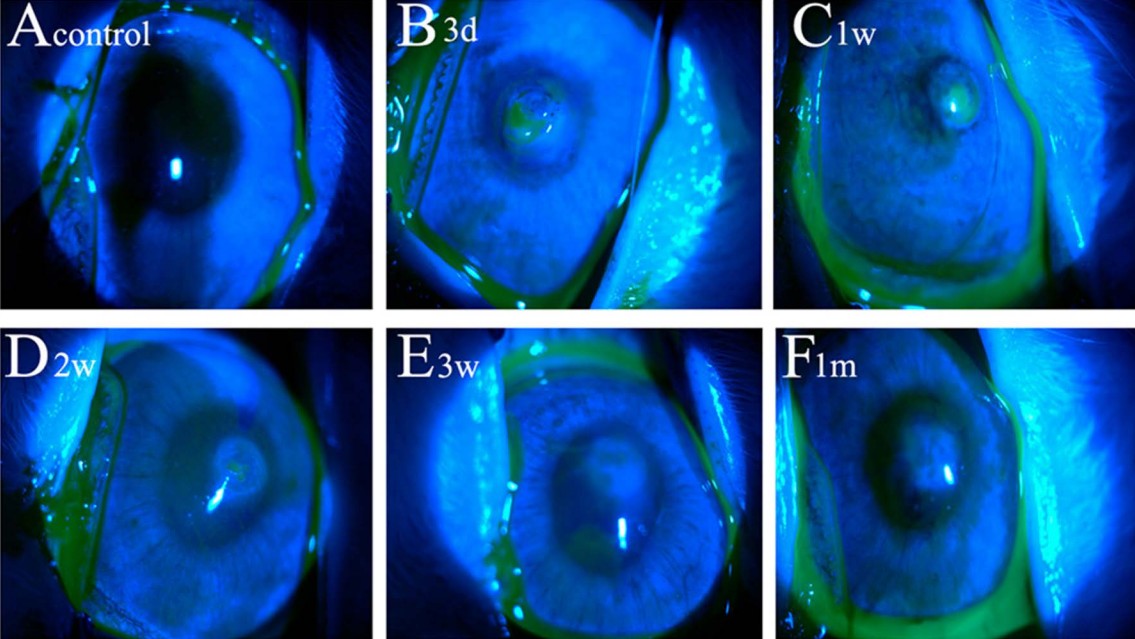

**Fig 1. Corneal epithelial defects after penetrating injury in each group. A**: normal control group, there was no fluorescein staining. **B**: At 3 days after injury, the corneal wound area was stained. **C**: At 1 week after injury, the staining area in the wound area decreased. **D**: At 2 weeks after injury, only a small area of fluorescein staining. **E**: At 3 weeks after injury, there was no fluorescein staining. **F**: At 1 month after injury, there was no staining.

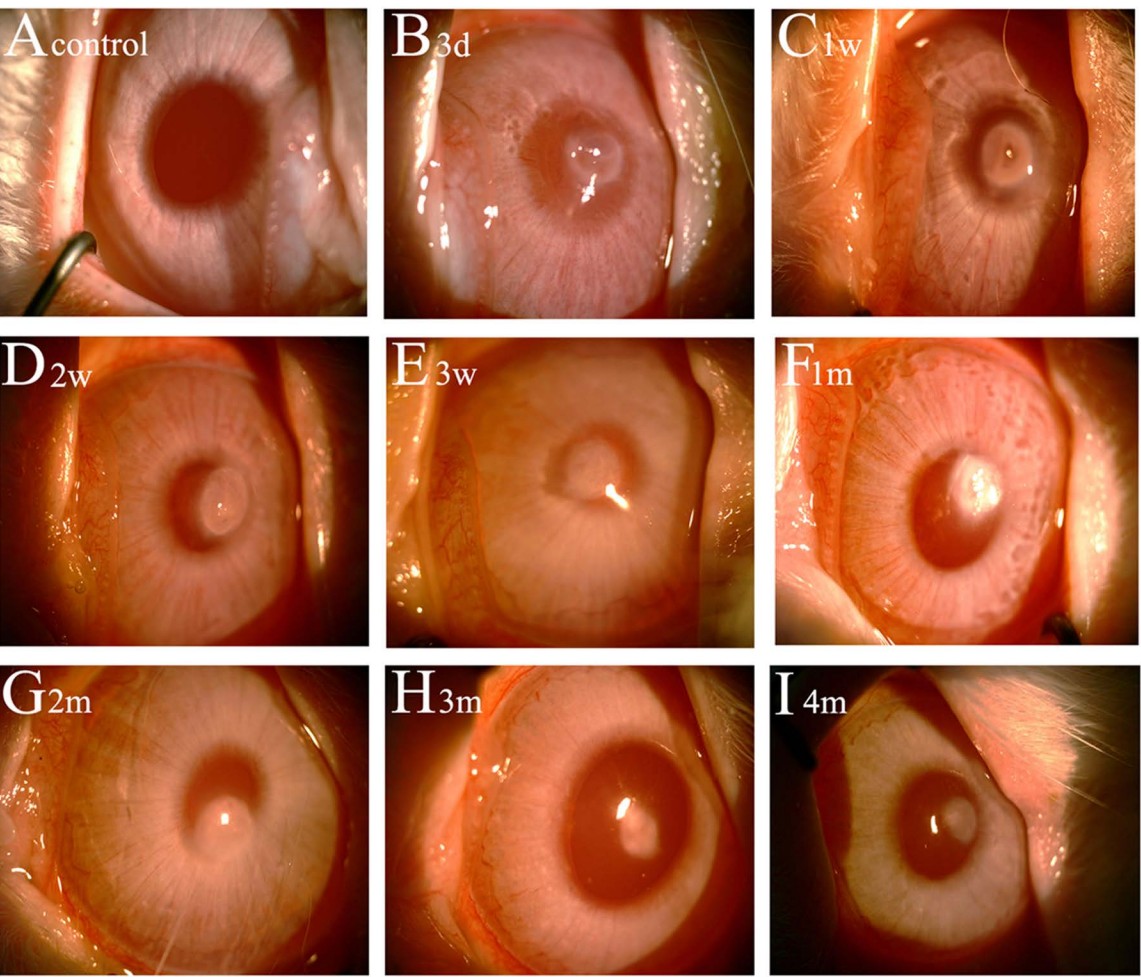

**Fig 2. Corneal opacity under slit lamp microscopy in each experimental group.** After injury, corneal opacity appeared at 3 days, gradually became denser, and peaked at 1 month. Corneal opacity progressively subsided at 2 to 4 months and was stabilized at 3 to 4 months.

2I), the wound area remained translucent with a clearly visible iris. As depicted in Fig 3A, complete epithelial healing was achieved at 3 weeks. Compared to the control group, the area of corneal epithelial defect exhibited significant differences at 3 days, 1 week, and 2 weeks post-injury (all $p < 0.05$). However, no significant difference was observed at 3 weeks and 1 month. Additionally, the corneal opacity scores progressively increased from 3 days to 1 month post-injury, followed by a gradual decrease from 2 to 4 months (Fig 3B). Significant differences were observed in the corneal opacity score at various time points following corneal penetrating injury. The score differed significantly from the control group at each time point (all $p < 0.05$).

## Histopathological changes after corneal penetrating injury

Histopathological changes in the corneal wound area were assessed using hematoxylin and eosin (H&E) staining (Fig 4). The findings revealed a progressive compaction of the corneal structure from the anterior to the posterior region throughout the corneal injury repair process. The control group exhibited a well-organized corneal structure with 5–6 epithelial cell layers,

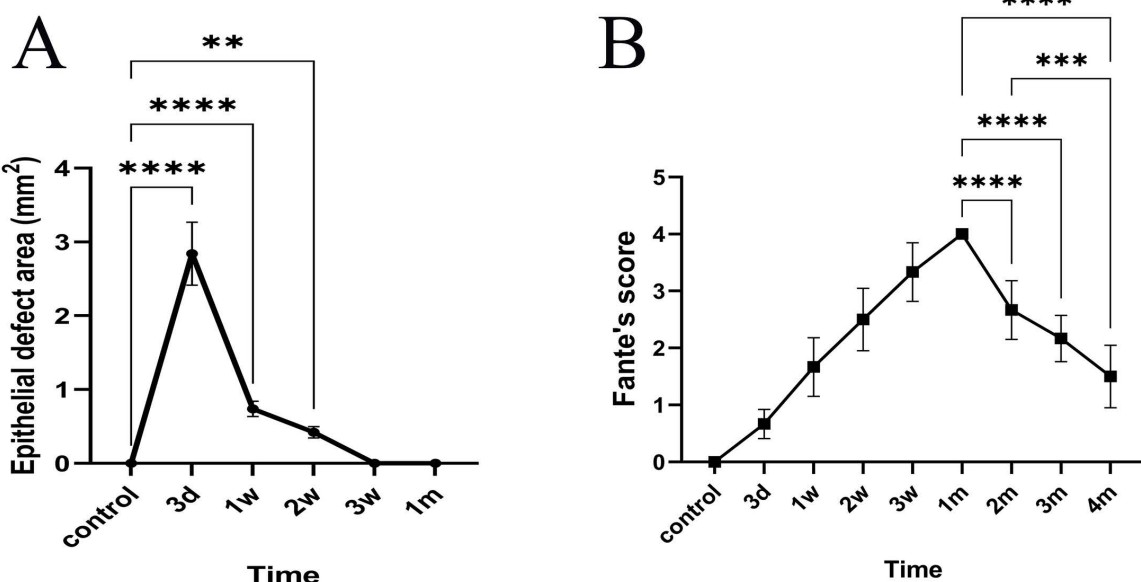

**Fig 3. Slit lamp examination after penetrating injury. A**: Assessment of corneal epithelial defect area. Compared with the control group, there were significant differences in the area of corneal epithelial defects at 3 days, 1 week, and 2 weeks after injury, but there was no significant difference at 3 weeks and 1 month after injury (**, $p \leq 0.01$; ****, $p \leq 0.0001$). **B**: Haze was evaluated by Fantes' score within 4 months. Corneal opacity progressively alleviated 2-4 months following penetrating injury, and the degree of corneal opacity was significantly lower at 4 months than at 1 month post-injury (***, $p \leq 0.001$; ****, $p \leq 0.0001$).

a compact stroma with regularly arranged collagen fibers, and a monolayer of endothelial cells parallel to the epithelium and stroma (Fig 4A). Three days post-injury (Fig 4B), the wound area was predominantly filled with acellular fibrous plaques, with sparse epithelial cell monolayers observed along the surface of fibrin clots. One week after injury (Fig 4C), the corneal epithelium displayed marked thickening and detachment from the stroma. The wound area exhibited a disorganized ECM, a loose stromal structure, and an irregular collagen arrangement. By two weeks post-injury (Fig 4D), the corneal epithelium decreased in thickness, and most fibrin clots within the wound area had disappeared. However, the distribution of the ECM remained irregular and disorganized. Notably, the lack of firm adhesion between the corneal epithelium and stroma at 3–14 days post-injury suggests incomplete EBM regeneration. Three weeks after injury (Fig 4E), the corneal epithelium displayed a regular arrangement, and the collagen fibers in the wound area's anterior stroma exhibited a tight organization, while those in the posterior stroma remained loosely arranged. At 1 month post-injury (Fig 4F), a compact arrangement of collagen fibers was observed throughout the anterior stroma. Two months post-injury (Fig 4G), the collagen fibers in the posterior stroma of the wound area displayed irregular organization. However, by 3–4 months after injury (Fig 4H and 4I), the full-thickness stromal fibers within the wound area were neatly arranged and resembled those of the uninjured cornea. These findings suggest that prior to the complete anchoring of the corneal epithelium to the stroma, the full-thickness corneal structure exhibits looseness and is primarily composed of fibroblasts and disorganized ECM. Conversely, when the corneal epithelium adheres tightly to the stroma, a compact distribution of collagen fibers is observed in the anterior stroma. Notably, the corneal stromal structure exhibited a progressive increase in compactness from the anterior to the posterior region during corneal wound healing.

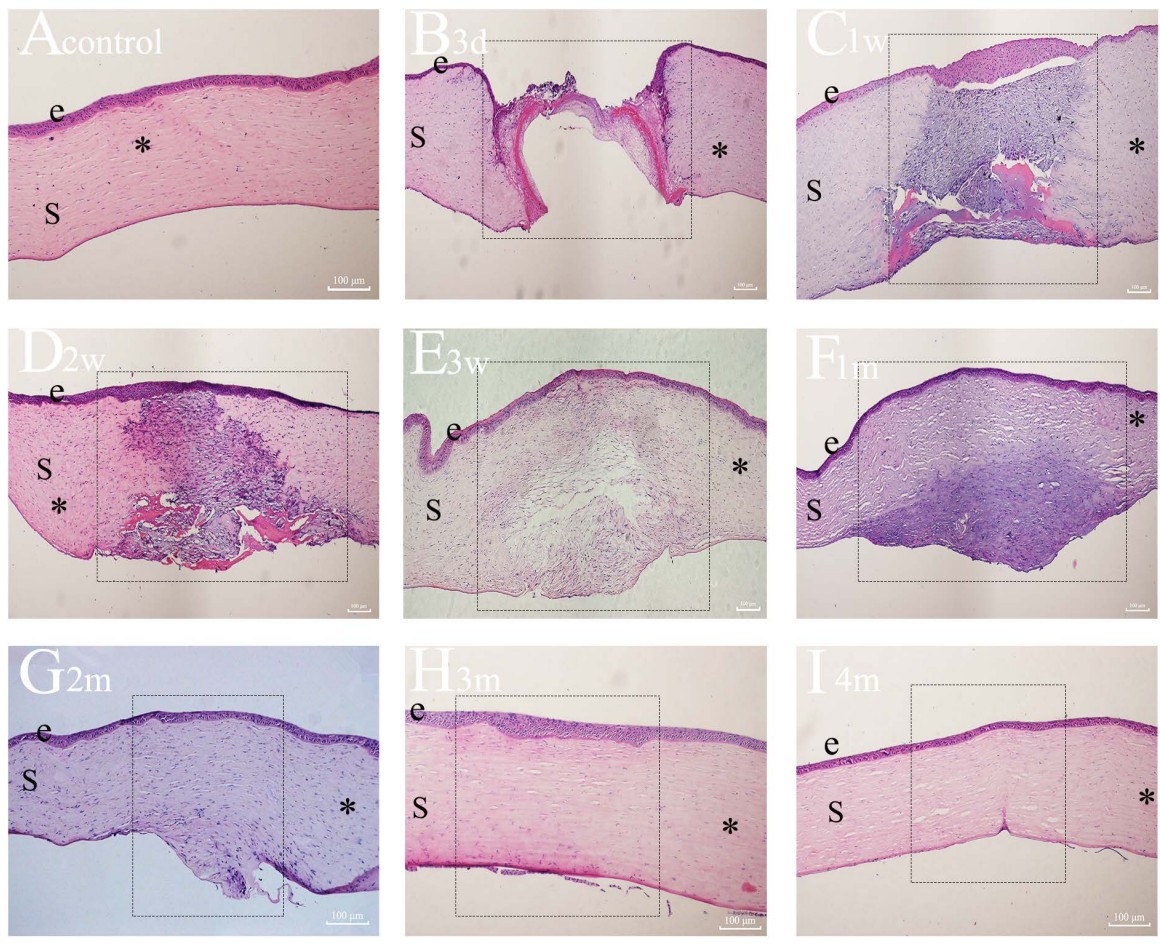

**Fig 4. Hematoxylin and eosin (H&E) stained sections of unwounded and penetrated corneas 3 days, 1 week, 2 weeks, 3 weeks, 1 month, 2 months, 3 months, and 4 months after modeling. e =epithelium, S=stroma, dotted rectangle=wounded area, *=unwounded area.** (Scale bars: 100 μm).

## Immunofluorescence staining for collagen types I and III

Immunofluorescence staining was employed to assess the distribution of collagen types I and III within the corneal wound area at each designated time point (Figs 5 and 6). The findings revealed that the reconstruction of collagen fibers in the anterior stroma preceded and surpassed that of the posterior stroma following full-thickness corneal injury. In the control cornea (Fig 5A), Col I exhibited a uniform distribution parallel to the epithelial and endothelial layers. Three days post-injury (Fig 5B), Col I expression was absent within the wound area. One week later (Fig 5C), Col I expression appeared throughout the wound area's full thickness in a disorganized manner, failing to reach the wound center. By two weeks post-injury (Fig 5D), Col I expression was significantly upregulated throughout the full-thickness stroma, particularly in the anterior region. However, the overall distribution remained disorganized. At three weeks (Fig 5E), a high density of Col I filled the entire stroma. While the distribution in the anterior 30–40% tended to be regular and parallel to the epithelium, the posterior 60–70% displayed a more disordered and intense expression compared to 2 weeks. At 1 month post-injury (Fig 5F), the anterior 50–60% of the stroma exhibited a regularly arranged

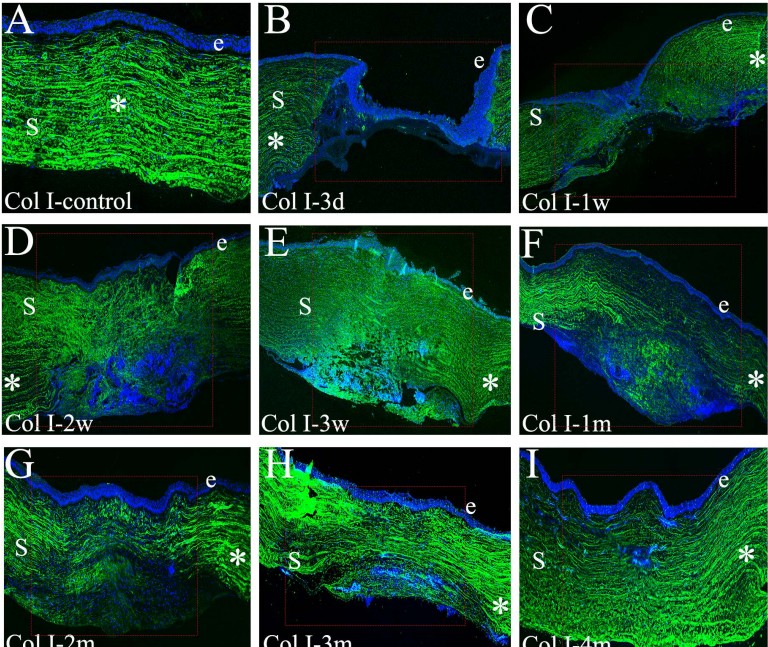

**Fig 5. Immunofluorescent staining for collagen type I expression and localization in unwounded corneas and wounded areas following penetrating injury.** In each panel, e represents the epithelium, S represents the stroma, while blue represents the DAPI staining of all nuclei. **A**: In the unwounded cornea, collagen type I (green) was regularly and uniformly distributed in the full-thickness stroma. **B**: Three days after injury, collagen type I was not detected in the corneal wound area. **C-D**: 1-2 weeks after injury, collagen type I was disordered in the whole stroma. **E-G**: At 3 weeks to 2 months after injury, collagen type I was regularly distributed in the anterior stroma. **H-I**: Immunohistochemical staining revealed a homogeneous distribution of collagen type I expression throughout most of the stromal region. The red dotted rectangle indicates the wounded area, and the asterisk (*) marks the unwounded area. (Scale bar: 100 μm).

and uniform distribution of Col I fibers, whereas the posterior half remained highly dense and disorganized. However, by 2 months post-injury (Fig 5G), Col I became regularly arranged in the anterior 60–70% of the stroma, with a tendency towards a more organized pattern in the remaining posterior region, albeit at a lower density. At 3 and 4 months post-injury (Fig 5H and 5I), similar to the unwounded cornea, Col I expression reached a high density throughout nearly the entire stromal thickness. However, a small portion of the posterior stroma still displayed disorganized collagen fibers. These findings suggest that while both anterior and posterior stroma undergo remodeling during corneal wound healing, the anterior region completes this process before the posterior stroma.

Immunofluorescence staining for Col III, a recognized marker of stromal fibrosis after corneal injury, revealed its absence in the normal cornea (Fig 6A). Similarly, Col III expression was undetectable within the wound area at 3 days post-injury (Fig 6B). However, by 1 week post-injury (Fig 6C), Col III expression spanned nearly the entire stromal thickness. This expression persisted throughout the full-thickness stroma at 2 weeks (Fig 6D). Subsequently, Col III began to disappear from the anterior stroma at 3 weeks post-injury, becoming predominantly localized within the posterior 60–70% (Fig 6E). At 1 month (Fig 6F), Col III remained absent in the anterior stroma but persisted at a high density within the posterior region. A further decrease in Col III expression was observed in the anterior and central stroma by 2 months post-injury, with its distribution restricted to approximately 30–40% of

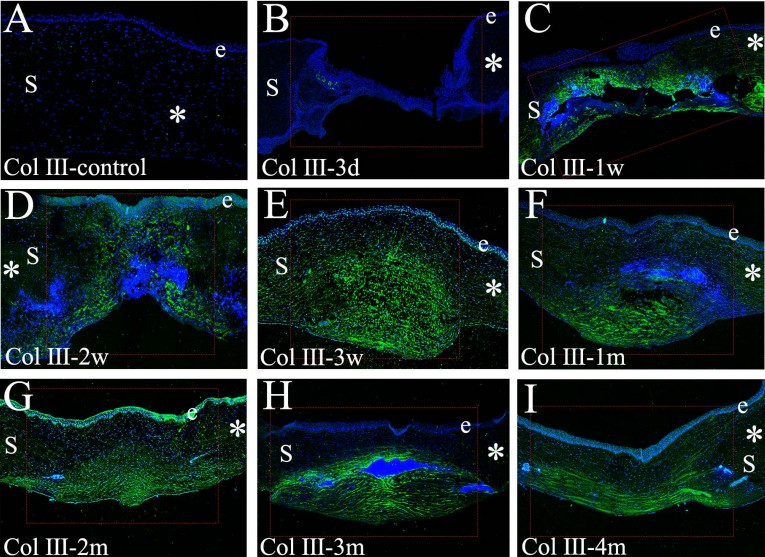

**Fig 6. Immunofluorescent staining for collagen type I expression and localization in control unwounded corneas and at specific time points after a penetrating injury in rabbits.** In each panel, e represents the epithelium, S represents the stroma, while blue represents the DAPI staining of all nuclei. In each instance, the depicted panel is representative of the results observed in two corneas at each time point per group. **A**: In the unwounded corneas, collagen type I was not detected in the wound area. **B**: At 3 days after injury, collagen type I was absent. **C-D**: At 1 to 2 weeks after penetrating injury, collagen type I (green) nearly filled the whole stromal thickness. **E-I**: At 3 weeks to 4 months after injury, collagen type I was observed in the posterior stroma. The red dotted rectangle=injured area, *=uninjured area. (Scale bar: 100 μm).

the posterior stroma (Fig 6G). Finally, Col III expression persisted within this posterior region (30–40%) at both 3 and 4 months (Fig 6H and 6I). These findings suggest that corneal stromal reconstruction involves a two-fold process: collagen fibril secretion originating from the wound periphery and a remodeling process that progresses from the anterior (proximal) to the posterior (distal) regions of the stroma. Additionally, they support the earlier observation that reconstruction of the anterior stroma precedes and surpasses that of the posterior stroma.

## Transmission electron microscopy for collagen fibers and basement membrane (BM)

TEM revealed normal EBM with integrated lamina lucida and lamina densa, as previously reported [28,29]. Fig 7 demonstrates that prior to EBM regeneration, the ECM was disorderly; following EBM regeneration, collagenous alignment became organized. In our study, TEM images of uninjured corneas revealed the normal corneal ultrastructure, including integrated lamina lucida and lamina densa of EBM, normal keratocytes positioned parallel to the epithelium, and a regular arrangement of collagenous fibers (Fig 7A). At 3 days post-(surgical modeling of a penetrating injury), an epithelium monolayer covered the corneal wound area, fibrovascular pannus filled the corneal stroma, and fibroblasts proliferated to generate reticular structures (Fig 7B). At 1 week after penetrating injury, collagenous fibers and myofibroblasts initially emerged in the stroma, the ECM was disordered, and there was no EBM (Fig 7C). Compared to 1 week after injury, collagen fibers and myofibroblasts significantly increased in the stroma at 2 weeks, and the ECM became more disorderly, while the EBM remained absent (Fig 7D). At 3 weeks after penetrating injury, the arrangement of collagenous fibers was irregular, myofibroblasts further increased in the stroma, and a large

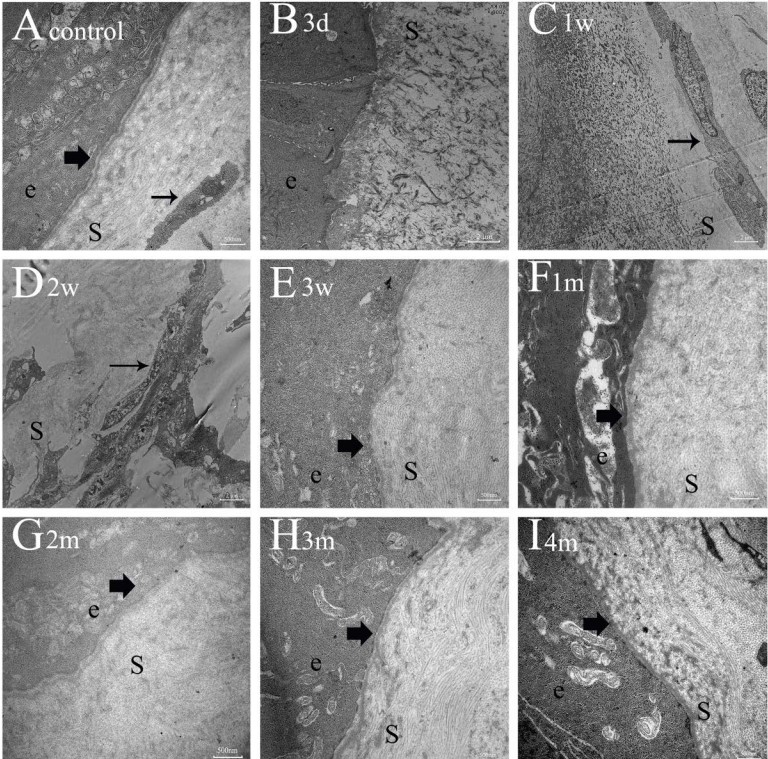

**Fig 7. Transmission electron microscopy (TEM) photographs. A**: Normal keratocytes were distributed parallel to the epithelium, collagen fibers formed lamellae, and the EBM had integrated lamina lucida and lamina densa (Magnification=30000). **B-D**: At 3 days to 2 weeks after injury, collagenous fibers and myofibroblasts significantly increased in the stroma, and the EBM remained absent (Magnification=7000). **E**: At 3 weeks after penetrating injury, collagenous fiber distribution was irregular and myofibroblasts increased further in the stroma while the defective EBM regenerated (Magnification=30000). **F-H**: One to 3 months after injury, collagenous fiber distribution became more and more regular, myofibroblasts decreased; the EBM was progressively restructured (Magnification=30000). **I**: Four months after penetrating injury, collagen fiber distribution was organized, the ECM was similar to the control group, myofibroblasts were practically undetectable, and the EBM had almost completely regenerated (Magnification=30000). e=epithelium, S=stroma, small arrow=myofibroblast, arrow=EBM.

proportion of ECM was disorganized; notably, the nascent EBM was defectively regenerated in a non-continuous manner (Fig 7E). One month after modeling, collagenous fibers reverted to a regular arrangement, myofibroblasts significantly decreased, the ECM was organized, and the EBM was continuous (Fig 7F). Two months after penetrating injury, collagenous fiber organization became increasingly regular, myofibroblasts decreased, the ECM was more organized, and the EBM gradually restructured and became more continuous compared to 1 month post-injury (Fig 7G). At 3 months and 4 months after injury, the distribution of collagenous fibers and ECM was similar to the control group, myofibroblasts were practically undetectable, and the EBM had regenerated to a nearly normal structure (Fig 7H and 7I).

Fig 8 demonstrates that remodeling of collagen fibers in the anterior stroma occurred earlier and more effectively compared to the posterior stroma. Additionally, Descemet's membrane exhibited incomplete regeneration following full-thickness corneal injury. At 2–4 months post-injury, corneal opacity progressively diminished under slit-lamp examination. To investigate this phenomenon, we examined TEM images to assess the gradual reconstruction of the corneal stroma during wound healing (Fig 8). The initial figure displays a healthy

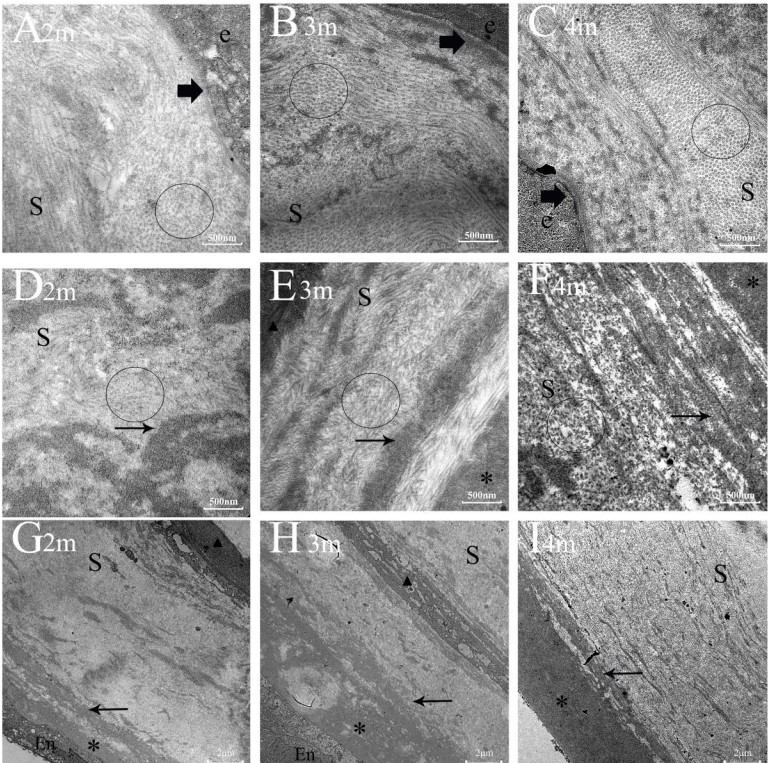

**Fig 8. Ultrastructure of the stromal collagen fibers and DM during the later stage of corneal wound healing.** Note that certain fibrils are observed longitudinally, whereas others are observed in cross-section; in the control group, all the fibrils have an identical diameter (Fig.8A). **A-C**: The remodeling of the diameter, arrangement, and orientation of collagen in the anterior stroma at 2-4 months after full-layer corneal injury. (Magnification=30000) **D-F**: The reconstruction of collagen in the posterior stroma at 2-4 months after corneal injury. (Magnification=30000) **G-I**: The regeneration of DM at 2-4 months after corneal perforation injury. (Magnification=7000) e=epithelium, En=endothelium, S=stroma, *=DM layer, small arrow=scattered DM, arrow=EBM, circle=cross-section of collagen fibers, ▲=myofibroblast.

cornea, characterized by a predominance of collagen fibers with a uniform, small diameter, and a highly organized lamellar arrangement within the stroma. These features are essential for corneal transparency. Two months after injury, the distribution of collagen fibers in the anterior stroma appeared disorganized. The shape and boundaries of individual fibers were somewhat discernible, but the cross-sections displayed irregularities, and the spacing between fibers was uneven (Fig 8A). The posterior stroma exhibited more pronounced disorganization of collagen fibers, making it impossible to distinguish their shapes and boundaries (Fig 8D). Furthermore, a defect was observed between the endothelial and stromal layers, along with evidence of scattered DM attempting to reform in the posterior stroma (Fig 8G). Compared to the two-month time point, collagen fibers at three months post-injury displayed a clearer lamellar orientation, distinct cross-sectional boundaries, and a tendency towards circular and elliptical shapes. The gaps between fibers also appeared smaller (Fig 8B). However, collagen fibers in the posterior stroma remained disorganized, with blurred shapes and boundaries. Their cross-sectional shapes and sizes were inconsistent, and the spacing between them was uneven (Fig 8E). The DM layer exhibited thickening, although a significant number of scattered DM components were still evident within the posterior stroma (Fig 8H). After four months, the distribution of collagen fibers in the anterior stroma had nearly returned to

normal. Individual fibers displayed circular or elliptical cross-sections with clearly defined boundaries. The spacing between fibers was uniform and regular (Fig 8C). While the collagen fiber distribution in the posterior stroma remained somewhat disorganized, the shapes and boundaries of individual fibers were becoming visible. However, their cross-sectional shapes were still irregular, and the spacing between fibers continued to be loose and uneven (Fig 8F). The DM layer appeared denser compared to three months, but a small amount of DM synthesis was still ongoing within the posterior stroma (Fig 8I). These findings suggest a gradual reconstruction of collagen fibers in the stroma following full-thickness corneal injury. Notably, this process occurred earlier and more effectively in the anterior stroma compared to the posterior stroma. Descemet's membrane regeneration, however, appeared incomplete.

## Discussion

Our previous study found that during the repair of corneal penetrating injury, α-SMA was a marker protein of myofibroblasts, appeared in the whole stroma at 3 days to 1 week after injury, disappeared in the anterior stroma at 3 weeks, was still present in the posterior stroma at 1 month, and could not be detected in the whole stroma at 2 months [11]. Over the years, several reports have shown that myofibroblast production is involved in corneal wound healing, and its overproduction is involved in corneal fibrosis [24,30,31]. Myofibroblasts are opaque and fill the wound area by producing a large number of disordered collagen fibers [1,2,32,33]. The corneal stroma exhibits a unique combination of features: small fibril diameter, parallel alignment, and regular interfibrillar spacing. These features contribute significantly to maintaining corneal transparency [2,7,34]. Type I and type III collagens play a critical role in scar formation. However, their excessive deposition is associated with corneal scarring [2,8,34]. Type I collagen constitutes approximately 90% of the protein content within the corneal stroma. Additionally, the past few decades have witnessed the identification of pathological type III collagen in corneal scar formation [2]. While collagen expression following corneal injury has been previously explored, few studies have investigated their specific roles in the wound healing process. Therefore, this study aimed to investigate the long-term changes associated with collagen reconstruction within the stroma. We also explored the relationship between BM regeneration and stromal remodeling over time. By gaining a deeper understanding of these changes in the context of corneal fibrosis development during wound healing, we hope to establish a theoretical foundation for identifying key factors influencing corneal fibrosis and guide the development of targeted therapies.

Limited information exists regarding the predominant location and key influencing factors associated with corneal scar formation following full-thickness injury. While previous studies have documented the persistence of scar tissue primarily within the posterior region of partially thickened corneas with lacerations at later stages [35], the specific mechanisms remain unclear. Research suggests that the disappearance of myofibroblasts from the anterior stroma by 2–4 months post-injury coincides with the regeneration of a normal EBM. Notably, at 4 months following keratitis affecting the central cornea (with damage to Descemet's membrane and endothelium), myofibroblasts can persist in 10–20% of the posterior stroma [36]. Medeiros et al. employed a rabbit model of mechanical endothelial injury to investigate the healing response within the posterior stroma [37]. Their findings revealed that posterior stromal cells undergo apoptosis in response to endothelial injury, mirroring the apoptotic response of anterior stromal cells to epithelial injury. This observation suggests a strong similarity between the healing processes in the anterior and posterior stroma.

This study provides hitherto undocumented pathological evidence that persistent fibrosis within the posterior stroma following full-thickness corneal injury can contribute to residual

corneal scarring. Our findings demonstrate that reconstruction of anterior stromal collagen fibers precedes and surpasses that of posterior stromal collagen fibers after a full-thickness injury. Notably, the stromal structure undergoes a transformation from a loose, whole-layer organization to a progressively more compact state, with this compaction commencing in the anterior stroma and progressing posteriorly. Initially, both type I and type III collagen exhibit a disorganized pattern throughout the wound area. However, type I collagen progressively adopts a more regular arrangement, starting anteriorly and extending posteriorly. In contrast, a small portion of the posterior stroma retains a disorganized collagen structure. Furthermore, type III collagen gradually disappeared from the anterior stroma towards the posterior stroma, persisting primarily in the latter region. Electron microscopy revealed consistently superior remodeling of anterior stromal collagen fibers compared to their posterior counterparts at corresponding time points. These findings collectively suggest that type III collagen is a key component of pathological corneal scars, and its persistence is likely linked to the continued presence of α-SMA in the posterior stroma. Following full-thickness corneal injury, corneal opacity reached its peak density at 1 month post-injury, followed by a gradual improvement. These observations further support the hypothesis that anterior corneal stroma reconstruction precedes and surpasses that of the posterior stroma and that persistent fibrosis within the posterior stroma is responsible for the persistence of corneal scars. From a pathological perspective, the persistence of corneal opacity in the late stages of full-thickness corneal injury appears to be primarily influenced by type III collagen within the posterior stroma. This collagen may disrupt the orthogonal lamellar arrangement of type I collagen, ultimately leading to decreased corneal transparency.

Several studies suggest a parallel between the roles of the EBM and DM in corneal injury healing. Marino et al. demonstrated in a *Pseudomonas aeruginosa*-infected rabbit keratitis model that EBM regeneration coincided with the disappearance of anterior stromal myofibroblasts within 2–4 months post-injury [36]. Similarly, a corneal endothelial injury model indicates that the DM plays a crucial role in regulating posterior corneal fibrosis, analogous to the EBM's regulatory role in anterior fibrosis [21]. Previous research has shown that in rabbit models of corneal alkali burns (typically induced by sodium hydroxide), severe injuries frequently damage both the corneal endothelium and Descemet's membrane [38]. This damage triggers myofibroblast formation and leads to excessive production of disorganized extracellular matrix by fibrotic stromal cells. Furthermore, persistent myofibroblasts impair the ability of these cells to promote basement membrane regeneration, thereby amplifying the corneal fibrotic response [29,39]. However, the independent healing responses of the EBM and DM following full-thickness corneal injuries remain poorly understood. Further investigations are necessary to elucidate their specific contributions to the overall healing process.

This study demonstrated that EBM regeneration following full-thickness corneal injury is both earlier and more complete compared to Descemet's membrane regeneration. Even with extended observation periods, DM regeneration remained defective. Defective and discontinuous EBM was initially observed at 3 weeks post-injury, progressively reconstructing to near-normal structure by 3–4 months. In contrast, dispersed DM was detected at 2 months and exhibited ongoing reorganization into the DM layer but remained structurally defective at 4 months. Interestingly, EBM regeneration coincided with self-remodeling of the anterior stroma, suggesting a potential cooperative role in stromal reconstruction. Conversely, defective DM regeneration persisted alongside posterior stromal fibrosis. This significant discrepancy between EBM and DM regeneration may be attributed to the characteristics of their adjacent cell types. The EBM, situated between the epithelial and anterior stromal cell layers, exhibits regeneration closely linked to epithelial repair [40,41]. Corneal epithelial cell proliferation and regeneration initiate EBM regeneration through the production of LAMA3,

a crucial early component of the EBM [29]. In our study, complete epithelial cell regeneration occurred 3 weeks post-injury, coinciding with the initiation of defective EBM regeneration. In contrast, the cells adjacent to the DM are the endothelial cells and posterior stromal cells. Endothelial cells possess limited regenerative capacity, primarily relying on expansion and migration [21,42]. Consequently, the repair of the damaged DM is primarily mediated by stromal cells. We hypothesize that the persistence of α-SMA in the posterior stroma reflects a wound-healing response. In other words, the persistence of posterior stromal fibrosis may result from ongoing stromal and DM repair within the wound area. Therefore, promoting the regenerative capacity of the endothelium after corneal injury may contribute to improved DM regeneration. This hypothesis warrants further investigation.

Following acute and severe corneal injury, the epithelium releases a variety of cytokines, including IL-1α. Keratocytes and corneal fibroblasts concurrently secrete chemokines that recruit bone marrow-derived cells into the corneal stroma [44]. These recruited cells include monocytes, macrophages, fibrocytes, and lymphocytes, many of which further amplify inflammatory and fibrotic responses through autocrine cytokine/chemokine production [41,44]. Within the cornea, the EBM and DBM critically regulate TGF-β1/2 diffusion into the stroma [14,45,46]. Damage to these membranes or their defective regeneration permits excessive TGF-β penetration. TGF-β1, TGF-β2, and PDGF collectively drive the differentiation of surviving keratocytes (those evading initial apoptosis) into vimentin-positive, keratocan-negative corneal fibroblasts [14,20,24]. These fibroblasts, in conjunction with limbal blood vessel-derived fibrocytes, undergo progressive differentiation into α-SMA-positive, desmin-positive, vimentin-positive, keratocan-negative myofibroblasts [43,47]. Myofibroblast differentiation from precursor cells is TGF-β-dependent, with PDGF acting synergistically to modulate the keratocyte-to-fibroblast and fibroblast-to-myofibroblast transitions [14]. As key collagen type I producers, myofibroblasts directly contribute to pathological extracellular matrix deposition. For instance, elevated TGF-β levels in fibrotic corneas correlate with excessive collagen accumulation and subsequent ECM disorganization [46]. PDGF further exacerbates this fibrotic phenotype by enhancing collagen synthesis [46,48]. TGF-β also disrupts basement membrane homeostasis. In epithelial-stromal injury models, TGF-β alters nidogen-1 distribution and function within the EBM [14]. Severe injuries (e.g., -9.0D PRK) result in aberrant incorporation of perlecan (a TGF-β-regulating BM component) into the EBM, while nidogen-1 accumulates abnormally in the anterior stroma surrounding myofibroblasts [14,19,24,49]. This imbalance suggests TGF-β-mediated dysregulation of BM composition during fibrosis. PDGF-nidogen interactions further modulate fibrotic progression. Both nidogen-1 and nidogen-2 in the EBM bind PDGF, a process essential for regulating keratocyte phenotypic transitions [14,50]. Injury-induced PDGF dysregulation likely impairs these interactions, promoting aberrant keratocyte behavior and fibrosis development [14,46].

The adult rabbit is a commonly employed model for investigating penetrating corneal injury. While the rabbit cornea shares a similar overall structure to the human cornea (epithelial layer, basement membrane, stroma, Descemet's membrane, endothelial layer), a key distinction lies in the absence of Bowman's layer in rabbits. Previous studies suggest that Bowman's layer may not play a significant barrier role, as various cytokines, growth factors, and molecules like EBM component perlecan readily traverse this layer in both healthy and wounded corneas [51]. However, species-specific variations exist in wound healing responses to corneal injury. Corneal healing in rabbits following PRK demonstrates similarities to human corneas. One notable difference is the significantly higher proportion of myofibroblast-derived fibrosis observed in rabbits with high-correction PRK. Additionally, the time course of fibrosis development and resolution in rabbits is typically more compressed compared to humans [51–53]. The regenerative capacity of the EBM has been demonstrated

in various animal models, including mice undergoing partial keratectomy, irregular photo-therapeutic keratectomy (PTK), and corneal alkali burn injuries [17,18,34]. Unfortunately, studies investigating BM regeneration in human corneas are scarce, likely due to the reliance on TEM for definitive identification of the characteristic BM layers, lamina lucida and lamina densa [28,29]. Currently, no technology exists for *in vivo* observation of human corneal BM regeneration.

In conclusion, our study suggests that persistent corneal opacity following full-thickness injury is likely attributable to the combined effects of sustained myofibroblast presence and aberrant ECM deposition within the posterior stroma. While EBM regeneration progresses alongside anterior stromal self-remodeling, Descemet's membrane exhibits a defective regenerative response, potentially reflecting a trauma-induced response of posterior stromal cells. This defective DM regeneration coincides with the persistence of posterior stromal fibrosis. Our findings further indicate that stromal cells actively produce collagen types I and III during full-thickness corneal injury repair. The observed disorganized arrangement of type I collagen and the persistence of type III collagen within the healing posterior stroma may contribute to the enduring corneal opacity.

## Author contributions

**Conceptualization:** Yuqing Luo, Luting Xie, Sumei Li, Shifang Xiao, Xia Li.

**Data curation:** Jingjing Chen, Yuqing Luo, Luting Xie.

**Formal analysis:** Jingjing Chen, Xia Li.

**Investigation:** Jingjing Chen.

**Methodology:** Jingjing Chen, Yuqing Luo, Luting Xie, Sumei Li, Shifang Xiao.

**Project administration:** Jingjing Chen, Yuqing Luo, Na Meng, Xia Li.

**Resources:** Luting Xie, Sumei Li, Shifang Xiao.

**Software:** Jingjing Chen, Na Meng.

**Supervision:** Xia Li.

**Validation:** Jingjing Chen, Na Meng.

**Visualization:** Jingjing Chen, Yuqing Luo.

**Writing – original draft:** Jingjing Chen.

**Writing – review & editing:** Jingjing Chen, Xia Li.

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
