## [Decision Letter · Decision Letter 0]

22 Nov 2024

PONE-D-24-34892Long-term alterations of collagen reconstruction and basement membrane regeneration after corneal full-thickness penetrating injury in rabbitsPLOS ONE

Dear Dr. Li,

Thank you for submitting your manuscript to PLOS ONE. After careful consideration, we feel that it has merit but does not fully meet PLOS ONE’s publication criteria as it currently stands. Therefore, we invite you to submit a revised version of the manuscript that addresses the points raised during the review process.

**The reviewer and EBM voiced some concerns about the paper that the authors are invited to address. The major points of concern are as follows:**

**1. It is unclear why the authors chose to study the alpha6 chain of type IV collagen. Also, the staining pattern does not match previous publications. This is a basement membrane component and yet, the sections show no staining of basement membranes. The used antibody is not certified for rabbit use by the manufacturer and the pattern appears to be non-specific. Besides, the authors do not disclose how did they fixed and processed the corneas, which may also be a confounding factor. The staining should be repeated with another antibody. The authors might like to use this one, as it is known to work with a number of species including the species of origin: https://www.chondrex.com/products/anti-human-alpha-iv-nc-antibody-clone-h-4**

**2. In the Introduction, please elaborate on the cytokines: TGFbeta, PDGF and how they participate in regulating the wound healing, myofibroblast differentiation, and fibrosis generation**

**3. Please provide details about surgery as suggested. Also, was pentobarbital used for anesthesia or euthanasia or both?**

**4. The authors do not really have any data to speculate that Col IVα6 plays a regulatory role in corneal fibrosis. It could be a secondary player, especially that the used antibody might have shown non-specific patterns. This statement is misleading and should be removed.**

**5. In Discussion, the authors could add previous data showing EBM and DM role in myofibroblast differentiation after alkali burn injuries in rabbits.**

**6. Please expand the discussion about the cytokine roles and basement membranes relations with TGFbeta and PDGF and myofibroblast differentiation.**

We look forward to receiving your revised manuscript.

Kind regards,

Alexander V Ljubimov, Ph.D.

Academic Editor

PLOS ONE

**Journal Requirements:**

Reviewers' comments:

Reviewer's Responses to Questions

**Comments to the Author**

1. Is the manuscript technically sound, and do the data support the conclusions?

Reviewer #1: Yes

2. Has the statistical analysis been performed appropriately and rigorously? 

Reviewer #1: Yes

3. Have the authors made all data underlying the findings in their manuscript fully available?

Reviewer #1: Yes

4. Is the manuscript presented in an intelligible fashion and written in standard English?

Reviewer #1: Yes

5. Review Comments to the Author

**Reviewer #1:**  Li and colleagues presented a very interesting research investigating the long-term alterations of collagen reconstruction and basement membrane (BM) regeneration after corneal full-thickness penetrating injury in rabbits.

Nice work and important contributions to the field.

A few relatively minor comments/questions/suggestions:

Introduction, Line 73: please describe more about the cytokines: TGFbeta, PDGF and how they participate regulating the wound healing and myofibroblasts differentiation and fibrosis generation

Material and Methods, line 91: please describe this surgery better, what was the size of injury? were suture places? any adverse events during or after surgery?

Material and methods, line 97-99 “General anesthesia was achieved through intravenous injection of pentobarbital sodium (3 ml/kg) via an auricular vein, followed by topical anesthesia with 0.5% proparacaine hydrochloride eye drops

Material and methods, Line 106-108: “Following general anesthesia, euthanasia was performed using an intravenous injection of 100 mg/kg pentobarbital sodium"

Was pentobarbital used for anesthesia ou euthanasia??

Please correct this information in the text

Discussion, line 422: Authors could add in the discussion previous data from papers that showed EBM and DM role and myofibroblasts differentiation after alkali chemical burn injuries in rabbits

Discussion, line 441 and 469: In rabbits the endothelium is also different - endothelium cells may regenerate in rabbits

Also please add in the discussion about the cytokines role - and basement membranes relation with TGFbeta and PDGF entrance and myofibroblast differentiation

Conclusion, line 496 / and abstract, line 39 (conclusion abstract) : “Our findings suggest that Col IVα6 plays a regulatory role in corneal fibrosis”—> with the information the paper gave it’s not possible to state this, better to use “appears to be involved in regulating corneal fibrosis” as used in the text.

6. PLOS authors have the option to publish the peer review history of their article (what does this mean? ). If published, this will include your full peer review and any attached files.

**Do you want your identity to be public for this peer review?** For information about this choice, including consent withdrawal, please see our Privacy Policy .

Reviewer #1: No

---

## [Author Response · Author response to Decision Letter 1]

8 Jan 2025

Dear Editors and Reviewers:

Thank you for your letter and for the reviewers' comments concerning our manuscript entitled“Long-term alterations of collagen reconstruction and basement membrane regeneration after corneal full-thickness penetrating injury in rabbits”(PONE-D-24-34892). Those comments are all valuable and very helpful for revising and improving our paper, as well as the important guiding significance to our researches. We have carefully considered your suggestions for improvement and will address each of the issues you raised. Revised portion are marked in the revised manuscript with track changes. The main corrections in the paper and the responds to the reviewer's comments are as flowing:

The reviewer and EBM voiced some concerns about the paper that the authors are invited to address. The major points of concern are as follows:

1. It is unclear why the authors chose to study the alpha6 chain of type IV collagen. Also, the staining pattern does not match previous publications. This is a basement membrane component and yet, the sections show no staining of basement membranes. The used antibody is not certified for rabbit use by the manufacturer and the pattern appears to be non-specific. Besides, the authors do not disclose how did they fixed and processed the corneas, which may also be a confounding factor. The staining should be repeated with another antibody. The authors might like to use this one, as it is known to work with a number of species including the species of origin: https://www.chondrex.com/products/anti-human-alpha-iv-nc-antibody-clone-h-4

Response: We appreciate your concerns about the alpha6 chain type IV collagen selection in this study and the specificity of the antibody. We have not noticed this before, which we will pay attention to in future experiments and verify with your recommended antibodies. In this revised manuscript, we have removed the results and conclusions related to collagen IV, which we believe have little impact on the final conclusions of the manuscript. At the same time, the methods you mentioned for fixing and processing the cornea are not disclosed, which may be a confusing factor. We have further disclosed the methods of corneal fixation in this revised manuscript. We will be happy to edit the text further, based on helpful comments from the EBM and reviewers.

2. In the Introduction, please elaborate on the cytokines: TGF-beta, PDGF and how they participate in regulating the wound healing, myofibroblast differentiation, and fibrosis generation

Response: We acknowledge your suggestion to highlight the pivotal role of cytokines in corneal wound healing, particularly the involvement of TGF-β and PDGF in myofibroblast differentiation during corneal fibrosis. As suggested, we have incorporated this emphasis into the introduction of the manuscript.

3. Please provide details about surgery as suggested. Also, was pentobarbital used for anesthesia or euthanasia or both?

Response: We recognize the importance of the operational details of the surgical procedure, and in the revised manuscript we have described the procedure in detail as recommended. Regarding the use of anesthetics, in particular pentobarbital, you ask whether it is used for anesthesia or euthanasia, or both. We can confirm that pentobarbital was used for both general anesthesia and euthanasia in our experiments. To ensure the welfare and safety of our animals, we strictly adhere to ethical guidelines and animal welfare protocols throughout our experiments.

4. The authors do not really have any data to speculate that Col IVα6 plays a regulatory role in corneal fibrosis. It could be a secondary player, especially that the used antibody might have shown non-specific patterns. This statement is misleading and should be removed.

Response: We acknowledge that in response to the data question you raised about whether Col IVα6 plays a regulatory role in corneal fibrosis, it is true that the current data may not be sufficient to fully support the speculations of this point of view. We also acknowledge that the staining patterns in the current experiment and the possibility of non-specific staining are factors we need to consider. The non-specificity of antibodies that you point out is an important consideration. We will pay more attention to the antibody selection and validation process in future studies to ensure that the results we get are reliable and scientifically based. With regard to the misleading statement you mentioned, we have deleted the relevant results and relevant misleading conclusions about type IV collagen in the revised manuscript.

5. In Discussion, the authors could add previous data showing EBM and DM role in myofibroblast differentiation after alkali burn injuries in rabbits.

Response: We acknowledge your suggestion to add a paragraph in the Discussion summarizing the EBM and DM role in myofibroblast differentiation after alkali burn injuries in rabbits. In the revised manuscript, we will ensure the inclusion of such a paragraph to provide a clear overview of the roles of EBM and DM in myofibroblast differentiation following rabbit alkali burn..

6.Please expand the discussion about the cytokine roles and basement membranes relations with TGFbeta and PDGF and myofibroblast differentiation.

Response: We acknowledge and concur with your recommendation to broaden the discussion on the role of cytokines as well as the interplay between the basement membrane, TGF-β, PDGF, and myofibroblast differentiation. This expansion will facilitate a more in-depth exploration of these topics, thereby enhancing our understanding of the specific functions of cytokines and basement membranes in biological processes and their interactions with TGF-β, PDGF, and myofibroblast differentiation. We have accordingly expanded this section in the revised manuscript's discussion and conducted an extensive review of relevant scientific literature to provide a more comprehensive and substantiated analysis.

Reviewer #1: Li and colleagues presented a very interesting research investigating the long-term alterations of collagen reconstruction and basement membrane (BM) regeneration after corneal full-thickness penetrating injury in rabbits.

Nice work and important contributions to the field.

A few relatively minor comments/questions/suggestions:

1. Introduction, Line 73: please describe more about the cytokines: TGF-beta, PDGF and how they participate regulating the wound healing and myofibroblasts differentiation and fibrosis generation

Response: We recognize the importance of cytokines in corneal injury repair and fibrosis formation. In the revised manuscript, we will provide cytokines such as TGF-β and PDGF and their role in regulating wound healing, myofibroblast differentiation, and fibrosis generation.

2. Material and Methods, line 91: please describe this surgery better, what was the size of injury? were suture places? any adverse events during or after surgery?

Response: We appreciate your emphasis on offering a more detailed description of the procedure. In this revised manuscript, we have provided an exhaustive account of the procedure, including specifics on the incision size, the exact location of sutures, and any complications or adverse events that were encountered.

3. Material and methods, line 97-99 “General anesthesia was achieved through intravenous injection of pentobarbital sodium (3 ml/kg) via an auricular vein, followed by topical anesthesia with 0.5% proparacaine hydrochloride eye drops

Material and methods, Line 106-108: “Following general anesthesia, euthanasia was performed using an intravenous injection of 100 mg/kg pentobarbital sodium"

Was pentobarbital used for anesthesia ou euthanasia??

Please correct this information in the text

Response: We acknowledge the insufficient description of the pentobarbital in the manuscript. In the revised manuscript, we have provided a comprehensive explanation of the utilization of the pentobarbital for both general anesthesia and euthanasia.

4. Discussion, line 422: Authors could add in the discussion previous data from papers that showed EBM and DM role and myofibroblasts differentiation after alkali chemical burn injuries in rabbits

Response: We acknowledge the importance of adding to the Discussion the role of EBM and DM in rabbit alkali burn and myofibroblast differentiation. In the revised manuscript, we have presented in detail the previous literature investigating the effects of EBM and DBM on myofibroblast differentiation after alkali burn in rabbits.

5. Discussion, line 441 and 469: In rabbits the endothelium is also different - endothelium cells may regenerate in rabbits

Response: We acknowledge that rabbit corneal endothelial cells are different from human corneal endothelial cells and can regenerate.

6. Also please add in the discussion about the cytokines role and basement membranes relation with TGF-beta and PDGF entrance and myofibroblast differentiation

Response: We acknowledge your suggestion to add a paragraph about the cytokines role and basement membranes relation with TGF-beta and PDGF entrance and myofibroblast differentiation. In the revised manuscript, we have incorporated an in-depth discussion on these topics, elaborating on the influence of cytokines, the interaction between the basement membrane and TGF-beta and PDGF, and their impact on myofibroblast differentiation.

7. Conclusion, line 496 and abstract, line 39 (conclusion abstract) : “Our findings suggest that Col IVα6 plays a regulatory role in corneal fibrosis” with the information the paper gave it’s not possible to state this, better to use “appears to be involved in regulating corneal fibrosis” as used in the text.

Response: We acknowledge that the statement "Our findings suggest that Col IVα6 plays a regulatory role in corneal fibrosis" was misstated and we have removed the misrepresentation in the revised manuscript.

Your invaluable insights have significantly elevated the quality and clarity of our manuscript, infusing it with greater depth and precision. We are profoundly grateful for your time and expertise in meticulously reviewing our work.

---

## [Editor Report · Decision Letter 1]

6 Feb 2025

PONE-D-24-34892R1Long-term alterations of collagen reconstruction and basement membrane regeneration after corneal full-thickness penetrating injury in rabbitsPLOS ONE

Dear Dr. Li,

Thank you for submitting your manuscript to PLOS ONE. After careful consideration, we feel that it has merit but does not fully meet PLOS ONE’s publication criteria as it currently stands. Therefore, we invite you to submit a revised version of the manuscript that addresses the points raised during the review process. The comments have been addressed. The only remaining concern is about some incorrect statements. For instance, in the discussion, the authors state "Nidogen-2 and collagen type IV, which are 474 primarily not associated with the epithelial basement membrane and DM". This is wrong for both proteins; the cited nidogen-2 reference utilized an antibody that was not certified for use in rabbits; in human cornea it is in the basement membranes and stromal keratocytes. Type IV collagen is the main BM component. Please also thoroughly check grammar and spelling.

We look forward to receiving your revised manuscript.

Kind regards,

Alexander V Ljubimov, Ph.D.

Academic Editor

PLOS ONE

Journal Requirements:

Additional Editor Comments:

The comments have been addressed. The only remaining concern is about some incorrect statements. For instance, in the discussion, the authors state "Nidogen-2 and collagen type IV, which are 474 primarily not associated with the epithelial basement membrane and DM". This is wrong for both proteins; the cited nidogen-2 reference utilized an antibody that was not certified for use in rabbits; in human cornea it is in the basement membranes and stromal keratocytes. Type IV collagen is the main BM component. Please also thoroughly check grammar and spelling.

---

## [Author Response · Author response to Decision Letter 2]

9 Feb 2025

Dear Editors and Reviewers:

Thank you for your letter and for the reviewers' comments concerning our manuscript entitled“Long-term alterations of collagen reconstruction and basement membrane regeneration after corneal full-thickness penetrating injury in rabbits”(PONE-D-24-34892R1). We sincerely appreciate the editors and reviewers’ insightful comments and constructive suggestions. All modifications have been carefully incorporated into the revised manuscript to enhance clarity, accuracy, and scientific rigor. Below is our point-by-point response to each comment.

Additional Editor Comments: The comments have been addressed. The only remaining concern is about some incorrect statements. For instance, in the discussion, the authors state "Nidogen-2 and collagen type IV, which are 474 primarily not associated with the epithelial basement membrane and DM". This is wrong for both proteins; the cited nidogen-2 reference utilized an antibody that was not certified for use in rabbits; in human cornea it is in the basement membranes and stromal keratocytes. Type IV collagen is the main BM component. Please also thoroughly check grammar and spelling.

Response: We truly appreciate the editor's thoughtful feedback. We acknowledge the inaccuracies in the original statement and have revised them in the manuscript. We corrected the misstatement "Nidogen 2 and collagen type IV, which are 474 primarily unrelated to epithelial basement membranes and DM" and then checked and corrected all spelling and grammar.

Your invaluable insights have significantly elevated the quality and clarity of our manuscript, infusing it with greater depth and precision. We are profoundly grateful for your time and expertise in meticulously reviewing our work.

---

## [Editor Report · Decision Letter 2]

11 Feb 2025

PONE-D-24-34892R2Long-term alterations of collagen reconstruction and basement membrane regeneration after corneal full-thickness penetrating injury in rabbitsPLOS ONE

Dear Dr. Li,

Thank you for submitting your manuscript to PLOS ONE. After careful consideration, we feel that it has merit but does not fully meet PLOS ONE’s publication criteria as it currently stands. Therefore, we invite you to submit a revised version of the manuscript that addresses the points raised during the review process.

The authors appear to have misunderstood the cited papers on nidogen-2 and type IV collagen. They have corrected an erroneous statement in previous version with another erroneous one: “The matrix components in this model, including nidogen-2 (detected using an antibody not validated for rabbit tissue) and type IV collagen, exhibit altered localization patterns compared to their physiological distribution in human corneal basement membranes. Specifically, nidogen-2 in the fibrotic stroma may reflect non-specific antibody binding or pathological stromal deposition, while type IV collagen is abnormally organized and lacks integration with the native basement membrane structures. [29, 39] Consequently, this distinct composition hinders the regenerative capacity of stromal keratocytes and fibroblasts”. 1. The cited papers did not deal with type IV collagen at all. 2. It is unnecessary to specify that the antibody was not validated in the studied species. 3. It is unclear what the pattern alteration was. 4. Nidogen-2 (and nidogen-1) is expressed in the stroma, thus, it is unclear what pathology the authors refer to. 5. Again, type IV collagen statement is made up as no such component was studied in the cited papers. 6. The statement about “hindering the regenerative capacity of stromal keratocytes and fibroblasts” is purely speculative and is not supported by data. Overall, the quoted sentences should be altogether removed as misleading, and the authors more carefully read the references.

We look forward to receiving your revised manuscript.

Kind regards,

Alexander V Ljubimov, Ph.D.

Academic Editor

PLOS ONE
---

## [Author Response · Author response to Decision Letter 3]

17 Feb 2025

Dear Editors and Reviewers:

We sincerely appreciate the time and effort dedicated by the reviewers and the editorial team to evaluate our manuscript titled “Long-term alterations of collagen reconstruction and basement membrane regeneration after corneal full-thickness penetrating injury in rabbits”(PONE-D-24-34892R2). We have carefully addressed all comments raised during the second round of review. The revised manuscript incorporates all suggested changes, and detailed point-by-point responses are provided below.

Response to Academic Editor’s Comments:

Comments� The authors appear to have misunderstood the cited papers on nidogen-2 and type IV collagen. They have corrected an erroneous statement in previous version with another erroneous one: “The matrix components in this model, including nidogen-2 (detected using an antibody not validated for rabbit tissue) and type IV collagen, exhibit altered localization patterns compared to their physiological distribution in human corneal basement membranes. Specifically, nidogen-2 in the fibrotic stroma may reflect non-specific antibody binding or pathological stromal deposition, while type IV collagen is abnormally organized and lacks integration with the native basement membrane structures. [29, 39] Consequently, this distinct composition hinders the regenerative capacity of stromal keratocytes and fibroblasts”.

1. The cited papers did not deal with type IV collagen at all.

2. It is unnecessary to specify that the antibody was not validated in the studied species.

3. It is unclear what the pattern alteration was.

4. Nidogen-2 (and nidogen-1) is expressed in the stroma, thus, it is unclear what pathology the authors refer to.

5. Again, type IV collagen statement is made up as no such component was studied in the cited papers.

6. The statement about “hindering the regenerative capacity of stromal keratocytes and fibroblasts” is purely speculative and is not supported by data. Overall, the quoted sentences should be altogether removed as misleading, and the authors more carefully read the references.

Response: We sincerely appreciate the editor's suggestions. We acknowledge that some omissions may have occurred in our literature review, and we will ensure a more comprehensive analysis in future work. As suggested, we have removed the indicated sentence from the forth paragraph of the Discussion section. This deletion improves the conciseness of the argument without affecting the overall logic of the section.

We believe these revisions have significantly improved the manuscript and addressed all reviewers’ concerns. We are happy to provide further modifications if needed. Thank you again for your time and valuable feedback.

Yours sincerely,

Xia Li

MD, PHD

Department of Opthalmology, The First Affiliated Hospital of Guangxi Medical University, 6# Shuangyong Road, Nanning, Guangxi, China

E-mail: lixiagmu066@163.com

---

## [Editor Report · Decision Letter 3]

25 Feb 2025

Long-term alterations of collagen reconstruction and basement membrane regeneration after corneal full-thickness penetrating injury in rabbits

PONE-D-24-34892R3

Dear Dr. Li,

We’re pleased to inform you that your manuscript has been judged scientifically suitable for publication and will be formally accepted for publication once it meets all outstanding technical requirements.

Kind regards,

Alexander V Ljubimov, Ph.D.

Academic Editor

PLOS ONE
---

## [Editor Report · Acceptance letter]

PONE-D-24-34892R3

PLOS ONE

Dear Dr. Li,

I'm pleased to inform you that your manuscript has been deemed suitable for publication in PLOS ONE. Congratulations! Your manuscript is now being handed over to our production team.

Kind regards,

on behalf of

Dr. Alexander V Ljubimov

Academic Editor

PLOS ONE